# Pancreatectomy with Celiac Axis Resection and Reconstruction for Locally Advanced Pancreatic Cancer

**DOI:** 10.3390/cancers16234115

**Published:** 2024-12-08

**Authors:** Satoshi Mizutani, Nobuhiko Taniai, Makoto Sukegawa, Takahiro Haruna, Hiroyasu Furuki, Hideyuki Takata, Junji Ueda, Masato Yoshioka, Takayuki Aimoto, Shunichiro Sakamoto, Kenji Suzuki, Yoshiharu Nakamura, Hiroshi Yoshida

**Affiliations:** 1Digestive Surgery, Nippon Medical School Musashikosugi Hospital, 1-383 Kosugimachi, Nakahara, Kawasaki 211-8533, Kanagawa, Japan; taniain@nms.ac.jp (N.T.); s-makoto@nms.ac.jp (M.S.); t-haruna@nms.ac.jp (T.H.); s2002083fh@nms.ac.jp (H.F.); hide-0323@nms.ac.jp (H.T.); junji0821@nms.ac.jp (J.U.); y-masato@nms.ac.jp (M.Y.); aimoto@nms.ac.jp (T.A.); 2Department of Cardiovascular Surgery, Nippon Medical School Musashikosugi Hospital, 1-383 Kosugimachi, Nakahara, Kawasaki 211-8533, Kanagawa, Japan; saka-165@nms.ac.jp (S.S.); suzuki@nms.ac.jp (K.S.); 3Department of Surgery, Nippon Medical School Chiba Hokusoh Hospital, 1715 Kamakari, Inzai 270-1694, Chiba, Japan; keishun@nms.ac.jp; 4Department of Surgery, Nippon Medical School, 1-1-5 Sendagi, Bunkyo, Tokyo 113-8603, Japan; hiroshiy@nms.ac.jp

**Keywords:** pancreatic ductal adenocarcinoma (PDAC), extended pancreatectomy, vascular resection and reconstruction

## Abstract

With the advent of effective chemotherapy, conversion surgery (CS) has been performed in patients who have responded to pretreatment, even for pancreatic cancer diagnosed as unresectable (UR) at the time of initial diagnosis. In CS, major arterial resection and reconstruction are necessary for complete radical resection. Many patients who require celiac axis (CA) resection combined with reconstruction have large tumors, poor findings of spread to surrounding tissues, and tumor invasion near the Abdominal Aorta; therefore, ingenuity is required for a safe resection. Furthermore, CA resection combined with reconstruction after specimen removal requires skills and experience in vascular surgery, such as selecting the arterial anastomosis site and the vessels to be used for bypass. We discuss the key points for safely performing pancreatectomy with CA resection combined with reconstruction, divided into resection (how to create “golden view”) and arterial reconstruction.

## 1. Introduction

Pancreatic cancer is a biologically highly malignant disease that easily leads to lymph node metastasis, nerve invasion, and distant metastasis even when the tumor is small [1]. Since Fortner [2] reported regional pancreatectomy in 1973 in an effort to improve surgical treatment outcomes, prophylactic extended surgery has been attempted worldwide to improve radical operation. Although some cases have benefited from this procedure, the 5-year survival rate after resection was less than 15%. It was difficult to improve the prognosis with surgery alone. Since Burris et al. [3] reported the superiority of chemotherapy with gemcitabine hydrochloride alone for unresectable and recurrent pancreatic cancer in 1997, several effective chemotherapy drugs have been published. Furthermore, the development of multi-drug combination therapy has progressed, and fluorouracil and calcium folinate combination therapy (FOLFIRINOX therapy) [4] and gemcitabine hydrochloride + nab-paclitaxel combination therapy [5] have become effective treatments for unresectable and recurrent pancreatic cancer.

In recent years, with the advent of effective chemotherapy [4,5,6,7], conversion surgery has been performed in patients who have responded to pretreatment, even for pancreatic cancer diagnosed as unresectable (UR) at the time of initial diagnosis. Reports from various studies have also elucidated factors associated with prolonged prognosis in conversion surgery (CS) [8,9,10].

However, for UR pancreatic cancer that has invaded the celiac axis (CA) or superior mesenteric artery (SMA) at the time of initial diagnosis, pancreatectomy combined with major arterial resection and reconstruction is often required, even with successful prior treatment. Major arterial resection is considered for cancer invasion into either or both arteries of the CA and SMA; however, the Japanese Clinical Practice Guidelines for Pancreatic Cancer weakly recommend only CA combined resection pancreatectomy [11].

Even when CS is possible, radical resection (R0 resection) requires major arterial resection and reconstruction, except when the patient has been downstaged from T4 to T3 on diagnostic imaging by pretreatment.

Many patients who require CA resection combined with reconstruction have large tumors, poor findings of spread to surrounding tissues, and tumor invasion near the Abdominal Aorta; therefore, ingenuity is required for a safe resection. Furthermore, CA resection combined with reconstruction after specimen removal requires skills and experience in vascular surgery, such as selecting the arterial anastomosis site and the vessels to be used for bypass.

In this study, we present the classification of progression of pancreatic cancer requiring pancreatectomy combined with arterial resection and reconstruction. Subsequently, we will discuss the key points for safely performing pancreatectomy combined with CA resection and reconstruction. Furthermore, we will include the treatment results from our hospital.

### Classification of CA Resection and Reconstruction Pancreatectomy

Pancreatectomy requires concurrent CA resection for arterial reconstruction. To begin with, most cases of distal pancreatectomy requiring concurrent CA resection are T4 pancreatic cancers centered in the pancreatic body (Figure 1a,b) [12]. Distal pancreatectomy with CA resection (DP-CAR) and gastrectomy-distal pancreatectomy-splenectomy (Appleby procedure) are commonly performed surgical procedures [13,14,15]. These procedures require concurrent CA resection but not arterial reconstruction; therefore, they have lower complication rates than techniques involving arterial reconstruction [14,16].

In contrast, in cases of pancreatic cancer with invasion of the root beyond the bifurcation of the common hepatic artery (CHA) and splenic artery (SpA) that cannot be radically resected with DP-CAR, CA resection combined with arterial reconstruction is required. Specific techniques include pancreaticoduodenectomy with CA resection (PD-CAR), total pancreatectomy with CA resection and total gastrectomy (TP-CAR+TG), and total pancreatectomy with CA resection (Spleen preserving) (TP-CAR[Spleen preserving]) (Figure 1c).

In these procedures, reconstruction of the hepatic artery (HA) and SpA or left gastric artery (LGGA) is required to maintain hepatic blood flow to the residual stomach, pancreas, and spleen. Graft reconstruction using the great saphenous vein is effective. As expected, postoperative bleeding in reconstructed arteries and serious complications related to arterial patency can occur.

Cases involving invasion of the CHA and tumor displacement into the CA are also classified as unresectable locally advanced (URLA) cancers; however, if the CA can be detached from the tumor, surgery can be performed by resection and reconstruction of the CHA alone. The procedure is pancreaticoduodenectomy with common hepatic artery resection (PD-CHAR) (Figure 1d).

## 2. Materials and Methods

### 2.1. Golden View

The possibility of safe pancreatectomy concurrent with CA resection and reconstruction depends on the ability to create a “golden view” that provides an unimpaired view of the Abdominal Aorta (Ao), CA, SMA, inferior vena cava (IVC), and left renal vein (LRV) from the ventral side (Figure 2a–c). Upon creating the golden view, it is possible to safely dissect the nerve plexus in the deepest part of the tumor, ganglia, and lymph nodes near the root of the CA and SMA. Furthermore, damage to the inferior transverse artery, right and left renal arteries, and other deeply located vessels that branch directly from the Ao can be avoided. Golden view serves as the basis for the visual field exposure necessary for pancreatectomy requiring CA reconstruction and for performing DP-CAR safely.

The extended Kocher mobilization and the mesenteric approach from the left and right sides to the anterior Ao, CA, and left side of the SMA are effective in creating the golden view. The Kocher mobilization is excellent for visual field exposure from the patient’s right side near the CA and SMA roots. Furthermore, the mesenteric approach is excellent for visual field exposure near the CA and SMA roots from the patient’s ventral and left sides.

### 2.2. Creation of the Golden View

#### 2.2.1. Kocher Mobilization

After laparotomy, the absence of liver metastasis or peritoneal dissemination should be confirmed. Intraoperative rapid cytodiagnosis should be performed to confirm that ascites lavage cytodiagnosis is negative for cancer. The omental bursa should be opened, and the absence of peritoneal dissemination should be confirmed. Simultaneously, the superior mesenteric vein should be detached as much as possible. Next, Kocher mobilization is performed. The layer exposing the anterior surface of the IVC and the left lateral margin should be dissected to detach to the left side of the Aorta. Extended Kocher mobilization is crucial for ensuring a clear visual field for subsequent CA and SMA root manipulations. At this point, the left adrenal vein should be ligated and dissected if possible. The right side of the celiac plexus, the first part of the pancreatic head plexus, and the SMA plexus should be dissected from the dorsal side to reach the SMA adventitia. The SMA celiac plexus, pancreatic head plexus, and SMA plexus are dissected dorsally using the double-flap technique for approximately 7 cm until the inferior pancreaticoduodenal artery is reached, exposing the adventitia. (Figure 3a). If the SMA can be separated from the tumor mass, the area surrounding the SMA is considered to be radically resectable.

Next, the right celiac ganglion and right arcuate ligament are cut to expose the anterior aspect of the Aorta. The root of the CA is dissected at the layer where the adventitia is exposed, while the celiac ganglion and celiac plexus are dissected from right to left (Figure 3b). If the root of the CA can be dissected, radical resection should be considered in the surrounding area of the CA, and a radical surgery should be initiated.

#### 2.2.2. Nakao Mesenteric Approach

The mesenteric root of the transverse colon is extensively dissected from the third part of the duodenum to the splenic flexure of the colon by a mesenteric approach using the Nakao method [17,18,19]. The superior mesenteric vein (SMV) and SMA are taped, and all other tissues are ligated and detached, including the inferior mesocolic vein (IMV). The third portion of the duodenum is the caudal resection margin. The middle colic artery (MCA) root is ligated and dissected to expose the mesenteric root of the transverse colon. Ligation and dissection of the MCA removes fixation of the transverse colonic mesentery from the retroperitoneum, allowing observation of the entire length of the SMA from the third portion of the duodenum to the root from the patient’s ventral side (Figure 3c). Next, the Treiz ligament on the patient’s left side of the SMA, the Tolds fusion fascia contiguous to its left side, and the Para-aortic lymph node are dissected on the resection side en bloc. If the left adrenal vein can be ligated and dissected at the time of Kocher mobilization, combined resection of the left adrenal gland can be easily performed.

#### 2.2.3. PD-CAR and TP-CAR(Spleen Preserving)

The procedure described up to this point is the same regardless of whether DP-CAR, PD-CAR, TP-CAR(spleen preserving), or TP-CAR+TG is used. In the subsequent procedure, there is a difference between PD-CAR and TP-CAR(spleen preserving), in which the pancreatic tail, spleen, and stomach are preserved, and TP-CAR+TG, in which all the aforementioned organs are resected en bloc.

In PD-CAR, the pancreas on the patient’s left side of the SMA is dissected, whereas in TP-CAR(spleen preserving), the entire pancreatic body and tail toward the side of the resected specimen is dissected, preserving only the SpA. In both procedures, the splenic vein is ligated and dissected at the same site. The dissection is continued dorsally, and the SMA plexus is further dissected from the patient’s ventral side, which is exposed in a double-door manner by Kocher mobilization, to dissect the SMA plexus in its entirety. The left celiac plexus and left celiac ganglion are further dissected to reach the anterior aspect of the Abdominal Aorta. Once the surgical process is completed, a golden view of the Ao, CA, SMA, IVC, and LRV can be seen from the ventral side (Figure 2b). When the golden view is completed, the resection specimen is connected to the patient’s body by the CA, SpA, proper hepatic artery (PHA), portal vein, and SMV. Subsequently, portal vein (PV) resection and reconstruction should be performed during Kocher mobilization. If the PV is not sufficiently long, the right external iliac vein is used as a graft. Finally, the CA, SpA, and PHA should be ligated and dissected after blood flow is blocked using clamping forceps. The specimen is removed as shown in Figure 3d.

#### 2.2.4. TP-CAR+TG

The surgical technique of TP-CAR+TG, in which the entire pancreas, spleen, and stomach are resected en bloc, does not preserve the SpA that was preserved by TP-CA (spleen preserving), and as per the surgical technique of radical antegrade modular pancreaticosplenectomy, the left celiac plexus and left celiac ganglion are dissected to reach the anterior aspect of the Ao [19]. After confirming the LRV and left renal artery, retroperitoneal tissue is detached en bloc along with the left adrenal gland, pancreatic body and tail, and spleen [20]. The esophagus is transected in the abdominal region.

Once the surgical process is completed, a golden view of the Ao, CA, SMA, IVC, and LRV can be obtained from the ventral side in TP-CAR+TG (Figure 2c).

At this point, the specimen is connected to the patient’s body via CA, PHA, PV, and SMV. PV resection and reconstruction are performed similar to those in PD-CAR.

Finally, the CA and PHA are dissected after blocking the blood flow using clamping forceps. The specimen is removed (Figure 3e).

### 2.3. CA Resection and Reconstruction

#### 2.3.1. Venous Bypass Graft

Pancreatectomy concurrent with CA resection requires arterial reconstruction, except for DP-CAR. Postoperatively, arterial blood flow must be maintained to the liver and remnant pancreas, stomach, and spleen. Maintaining vascular patency is crucial. To achieve this, the following two conditions should be observed: (1) tension-free and (2) short bypass. Graft bypass is essential to achieve (1) tension free. There are two types of grafts: biological grafts and artificial vessels. Gastrointestinal cancer surgery is classified as Class II (clean–contaminated surgery) in the Centers for Disease Control and Prevention (CDC) guidelines [21], and biological grafts are selected to prevent graft infection. Biological grafts include the choice of arteries and veins; however, it is virtually impossible to autoharvest an artery that is size-compatible with CAs > 5 mm in diameter. Therefore, the graft for anastomosing the CA should be a venous graft. In general, the great saphenous vein is the easiest to use in terms of diameter and length. With regard to (2), the shorter the graft length, the less likely it is to bend and twist and the more effective it is in preserving patency.

#### 2.3.2. Procedure for Vessel Anastomosis

##### CA Reconstruction in TP-CAR+TG

CA reconstruction in TP-CAR+TG involves anastomosis of CA and PHA or RHA, and if a CA branch length from Ao of approximately 15 mm can be ensured, CA–graft–PHA (RHA) end-to-end anastomosis is possible (Figure 4a). Everted continuous anastomosis is a reliable method using monofilament nonabsorbable threads as fine as 6-0 or 7-0. If the CA branch length from Ao is not sufficiently long for anastomosis, direct graft anastomosis to Ao is necessary.

##### CA Reconstruction in PD-CAR and TP-CAR(Spleen Preserving)

This procedure requires maintenance of arterial blood flow in the remnant pancreas, stomach, and spleen, and thus requires maintenance of blood flow in the SpA and/or LGA in addition to CA and PHA (RHA) anastomosis. If the CA branch length is relatively long, end-to-side anastomosis of CA–SpA and CA–LGA is possible (Figure 4b). If the CA branch length is not sufficiently long, an Ao–SpA anastomosis or Ao–LGA anastomosis is required. We performed CA–PHA and CA–SpA graft bypass using a Y graft, which is a modified saphenous vein graft (Figure 4c). However, this method is controversial from the perspective of “short bypass”, which is a condition for maintaining patency [22]. After the anastomosis is completed, the arterial blood flow status is checked by color doppler echocardiography.

After confirming that arterial blood flow is no longer a concern, pancreatojejunostomy, choledocojejunostomy, esophagojejunostomy, and gastrojejunostomy are performed, as appropriate for each procedure.

### 2.4. Patients

#### Indications

The indications for CS for URLA at our hospital are as follows: (1) patients aged <80 years, (2) patients with good performance status after pretreatment, (3) patients who underwent four or more courses of pretreatment, (4) patients with Stable Disease (SD) [23] or better response on imaging, and (5) CS should be a radical resection. Furthermore, cases involving simultaneous resection and reconstruction of CA and SMA are excluded. In addition, the indication for CA reconstruction is discussed with cardiovascular surgeons based on preoperative imaging findings to determine whether arterial bypass reconstruction is feasible.

## 3. Results

From 2014 to 2024, 16 patients were diagnosed with URLA pancreatic cancer at our hospital and underwent CS, requiring major artery en bloc resection after pretreatment. We performed DP-CAR in eight patients, gastrectomy-distal pancreatectomy-splenectomy (Appleby procedure) in one patient, PD-CHAR in two patients, PD-CAR in two patients, TP-CAR(spleen preserving) in one patient, and TP-CAR+TG in two patients. Caudal pancreatectomy, which did not require CA reconstruction, accounted for most cases. In total, five patients (31.3%) required surgery with CA reconstruction. All five patients required concurrent PV resection and reconstruction. All concurrent CA resection and reconstruction cases are presented in Table 1. Pretreatment included gemcitabine + S1 in two patients and gemcitabine + nab-paclitaxel in three patients.

The median pretreatment duration was 153 days. The pretreatment efficacy evaluation (response evaluation criteria for solid tumors: RECIST) was SD for 80%. The normalization of CA19-9 was 60%. The median operative time was 505 min, median blood loss was 950 g, and postoperative liver function tests and histopathological diagnoses are presented in Table 1. Four of the five patients had T4 pancreatic cancer. The R0 surgical rate was 80%. Complications of Clavien–Dindo (CD) IIIa or higher was observed in one patient (pneumonia; Case 4) (Table 1). In other patients, there were no perioperative complications, including pancreatic fistula. There were no deaths.

Although aspartate aminotransferase and alanine aminotransferase levels were elevated on the day following surgery, their levels improved on postoperative day 7 and 14, respectively, indicating that the effect of hepatic blood flow blockade was minimal. The patency of the graft artery was judged to be good on postoperative contrast-enhanced computed tomography CT (Figure 5a,b). Pneumonia was observed after the Clavien–Dindo IIIa procedure in Case 4. The length of hospital stay after surgery was generally good, except for 104 days in Case 4. All patients required antidiarrheal agents because of extensive upper abdominal plexus dissection. The histopathologic curability diagnosis of the resection specimen showed that in Case 3, the cancer cells had spread to the SMA plexus dissection surface, resulting in R1; however, R0 surgery could be performed in all the remaining cases (Table 1). In the histological evaluation after preoperative treatment according to the Evans classification [24], grade IIa (destruction of 10%–50% of tumor cells) was observed in three patients and grade I (little [<10%] or no evident tumor cell destruction) was found in two patients. Three patients were successfully treated with postoperative S1 chemotherapy within 6 weeks after surgery.

Regarding postoperative glycemic control in patients with TP, the combination of rapid-acting insulin and long-acting insulin controlled blood glucose in the range of 70–160 mg/dL. At 3 months after surgery, all patients experienced weight loss (8–13 kg). Despite the small number of cases, the median survival time was 24.5 months, which was not as good as the survival time reported by other studies showing the superiority of CS [25,26,27]; however, considering that most of the patients had histopathologically T4 pancreatic cancer, the pretreatment period was short, and the effect of the pretreatment was poor (SD accounting for 80% and 60% normalization ratio of CA19-9); more cases are necessary to discuss the pros and cons of treatment.

## 4. Discussion

The primary treatment for unresectable pancreatic cancer (UR) [28] is chemotherapy or chemoradiation, regardless of distant metastasis (URM) and local progression (URLA) [10,29,30]. Furthermore, with the advent of highly effective chemotherapy in recent years, curative surgery has become possible in some patients who were initially diagnosed with UR cancer at the time of initial diagnosis. CS is defined as primary tumor resection performed when radical resection (R0 resection) is feasible. With the advent of potent chemotherapy regimens such as FOLFIRINOX, a multidrug combination therapy including gemcitabine hydrochloride, reports began to emerge in the 2010s. Although there are many reports from Japan and abroad [10,31,32], the issues that must be addressed are (1) applicable patients, (2) regimens of chemotherapy and chemoradiotherapy and the duration of pretreatment (number of courses), (3) ensuring safety in highly invasive surgery (concurrent reconstruction of major arteries) and perioperative complications, and (4) verification of long-term prognosis of CS. As mentioned above, many reports have addressed (1) and (2), and further progress is expected in the future. Regarding (3), arterial reconstruction that ensures organ blood flow following concurrent major vessel resection is important for ensuring safety (concurrent major artery reconstruction) and perioperative complications in highly invasive surgery. Pancreatectomy requiring concurrent CA resection and reconstruction is performed to preserve hepatic, gastric, and splenic blood flow [31,32]. In contrast, pancreatic resection requiring concurrent SMA resection and reconstruction is performed to preserve blood flow in the small intestine, right lateral and transverse colon, and much of the intestinal tract [31,33]. Therefore, poor patency of arterial reconstruction can directly lead to a life-threatening condition. A Japanese study using the National Clinical Database examined 2167 cases of total pancreatectomy without concurrent CA resection and reconstruction performed between 2013 and 2017 [34]. Results showed a 30-day postoperative mortality rate of 1.0%, an in-hospital mortality rate of 2.7%, and the incidence of complications with a Clavien–Dindo classification of Grade III or higher was 6.0%. Compared with these data, the surgical outcomes at our hospital, where TP-CAR and TP-CAR+TG, accounting for most cases, can be judged to be generally favorable. With an emphasis on perioperative complications, pancreatectomy concurrent with CA resection and reconstruction should be performed in a high-volume center of hepatobiliary surgery that can collaborate with vascular surgery [33,35].

Regarding (4), the effect of preoperative treatment is a crucial factor for long-term prognosis after CS. The insignificant prolongation of long-term prognosis observed at our institution may be attributed to the fact that most of the postoperative histopathological diagnosis is T4 pancreatic cancer. In other words, surgery may be indicated in patients with an inadequate response to pretreatment [25,26,27]. Our study has several limitations associated with the inherent errors and biases of small studies. A large number of trials are recommended to further evaluate the feasibility of pancreatectomy requiring CA resection and reconstruction for URLA pancreatic cancer.

## 5. Conclusions

With the advent of anticancer drugs that are highly effective for the treatment of pancreatic cancer, there will be more opportunities to perform CS for patients with an initial diagnosis of URLA in the future.

Parallel to the resumption of pretreatment and the selection of suitable patients, surgeons must be prepared to safely and reliably perform pancreatectomies that require concurrent major arterial resection and reconstruction.

## Figures and Tables

**Figure 1 cancers-16-04115-f001:**
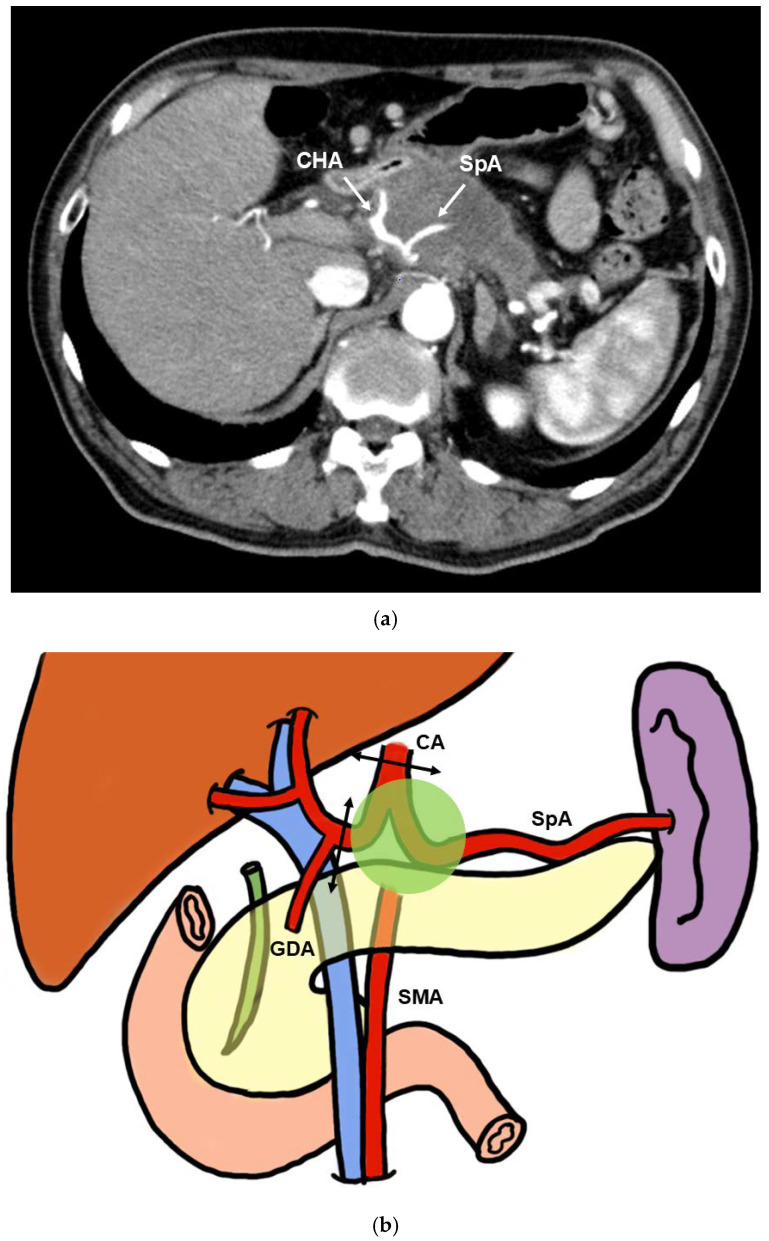
(**a**) Contrast computed tomography. CHA, SpA, and CA exhibit significant tumor invasion. (**b**) Extent of tumor invasion in the DP-CAR and Appleby procedures. CHA, SpA, and CA exhibit tumor invasion. The CA and CHA should be ligated and detached (arrows). (**c**) Extent of tumor invasion in PD-CAR, TP-CAR(spleen preserving), and TP-CAR+TG. In addition to CHA, SpA, and CA, the gastroduodenal artery (GDA), proper hepatic artery (PHA), and bile duct exhibit tumor infiltration. The CA and PHA should be ligated and detached (arrows). In PD-CAR and TP-CAR(spleen preserving), the SpA should be cut and reconstructed (dotted arrow). (**d**) Extent of tumor invasion in PD-CHAR. The CHA is infiltrated, but only the CA is displaced. The CHA of the area with tumor invasion is cut, and the severed ends are anastomosed end-to-end (arrows).

**Figure 2 cancers-16-04115-f002:**
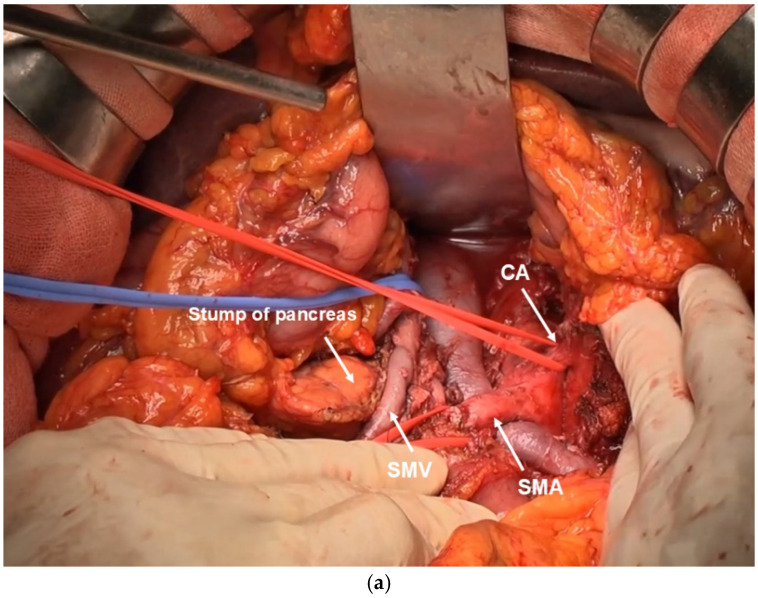
(**a**) Golden view in DP-CAR. The pancreatic stump is shown on the right side of the SMV. The tumor is moved to the left side along with the retroperitoneal tissue. The CA and SMA face opposite directions. (**b**) Golden view in TP-CAR+TG. The pancreatic body and tail, spleen, and stomach are moved to the right side en bloc (**). The CA and SMA face toward the same direction. (**c**) Golden view in PD-CAR. The dissected pancreatic surface and the remaining pancreatic body and tail can be seen on the left side. The CA and SMA face the same direction. The splenic vein was preserved on the splenic side, considering that it would drain to the IMV.

**Figure 3 cancers-16-04115-f003:**
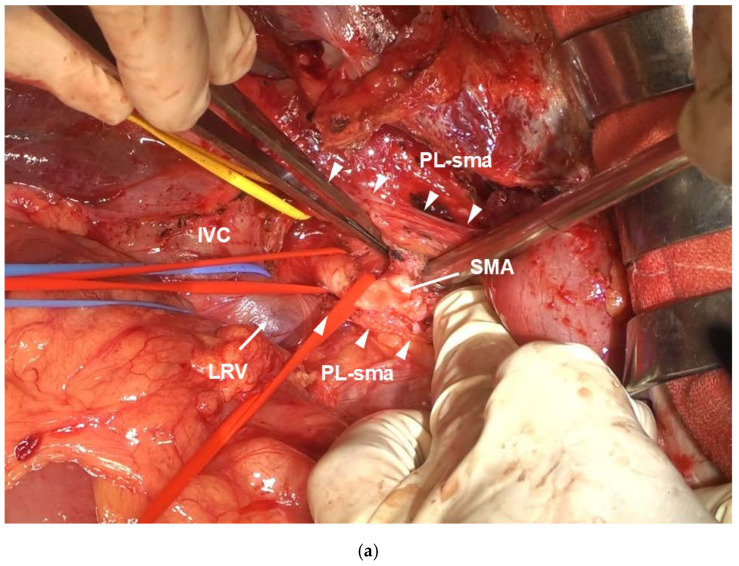
(**a**) In the visual field of Kocher mobilization, the SMA plexus (Δ) is opened dorsally in a double-door manner across approximately 7 cm from the SMA root. (**b**) The right celiac plexus and right celiac ganglion are separated to reach the anterior aspect of the Ao. (**c**) The mesenteric approach allows observation of the entire SMA length from the ventral side. (**d**) Photograph of the completed resection by DP-CAR. The CA stump is clamped with vascular clamping forceps, leaving a suture margin required for anastomosis. (**e**) Photograph of the completed resection using TP-CAR+TG. Wide field of view because of en bloc dissection of the left upper abdominal organs.

**Figure 4 cancers-16-04115-f004:**
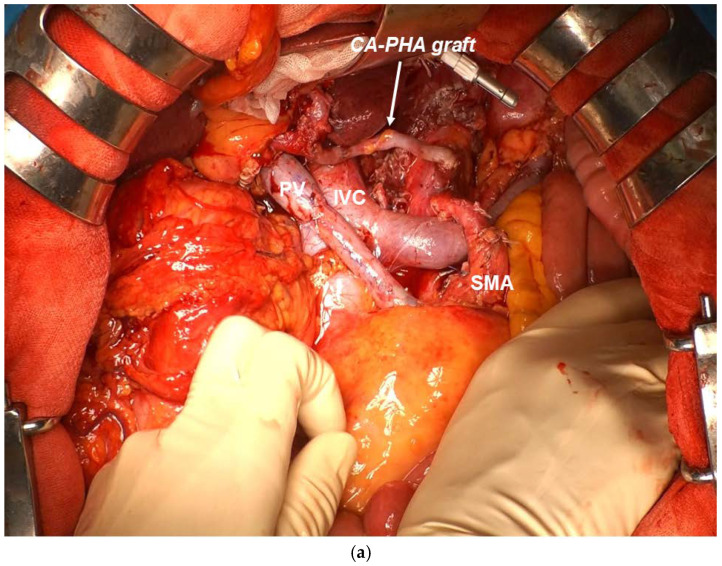
(**a**) CA–RHA anastomosis in TP-CAR+TG. CA–saphenous vein–RHA end-to-end anastomosis. (**b**) CA–RHA and CA–SpA anastomosis in TP-CAR(spleen preserving). CA–saphenous vein–RHA is connected with end-to-end anastomosis, and the respective SpA and LGA stumps and the CA lateral wall are anastomosed with end-to-side anastomosis. (**c**) CA–RHA and CA–SpA anastomosis in DP-CAR. CA–RHA and CA–SpA anastomosis with end-to-end anastomosis using a saphenous vein–modified Y graft.

**Figure 5 cancers-16-04115-f005:**
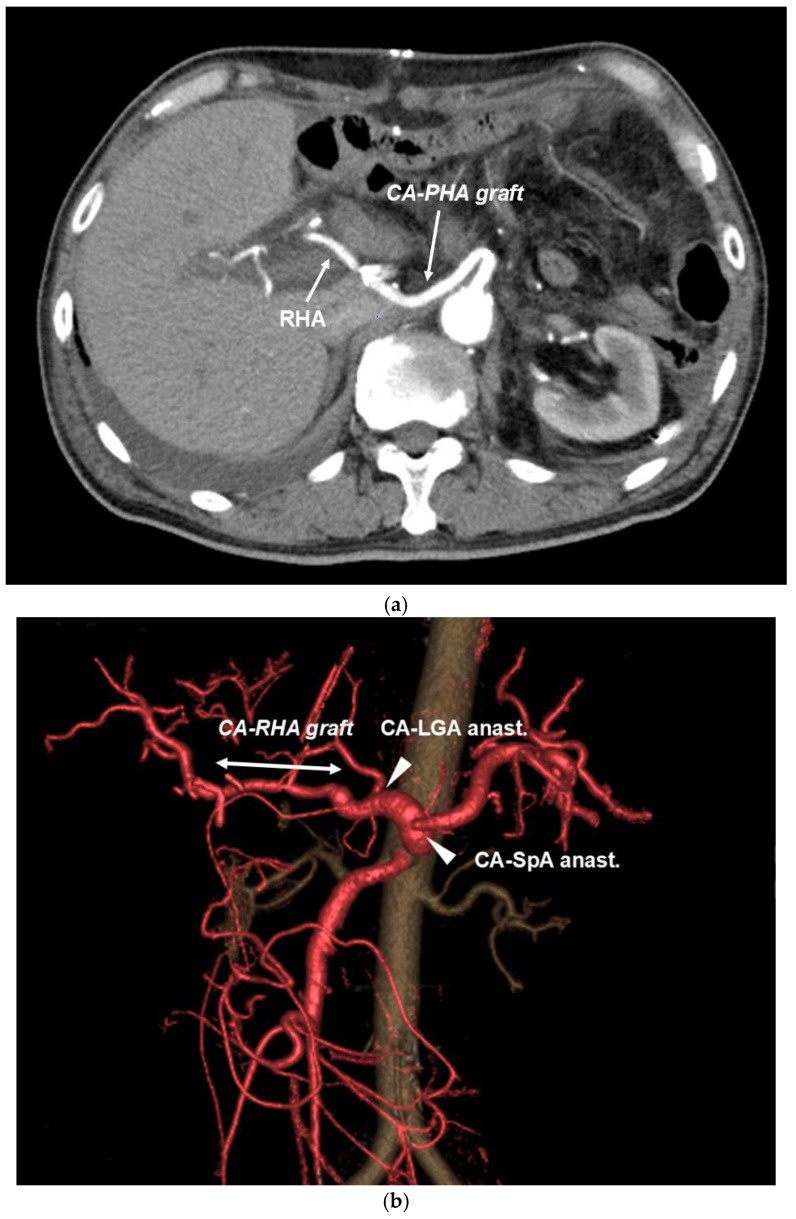
(**a**) TP-CAR+TG postoperative contrast CT. Good patency after end-to-end anastomosis artery reconstruction of CA–RHA with graft can be confirmed. (**b**) TP-CAR(spleen preserving) postoperative contrast 3DCT. End-to-end anastomosis of CA–RHA with graft and good patency after arterial reconstruction of LGA–CA and SpA–CA end-to-side anastomosis can be confirmed.

**Table 1 cancers-16-04115-t001:** Patient characteristics and surgical details.

Case	1	2	3	4	5
operation	PD-CAR	TP-CAR+TG	TP-CAR+TG	PD-CAR	TP-CAR(spleen pres.)
NST	GnP	GS	GS	GnP	GnP
period of NST(day)	125	153	142	226	255
CA19-9 normalization	yes	yes	no	no	yes
RECIST	SD	PR	SD	SD	SD
operation time(m)	505	345	732	451	522
blood loss	785	910	950	3117	2635
arterial reconstruction	CA-RHA	CA-PHA	CA-RHA	CA-PHAand SpA (Y-graft)	CA-RHA, CA-SpA, CA-LGA
venous reconstruction	PV-SMV	PV-SMV	PV-SMV	PV-SMV	PV-SMV
histological stage	T3	T4	T4	T4	T4
**invade artery**	CHA, SpA	CA, CHA, SpA	CA, CHA, SpA	CA, CHA, SpA	CA, CHA, SpA
**histopathology**	Mod > por	por	Mod > por	mod	Por > mod
**Evans classification**	lla	lla	l	l	lla
**Radicality**	R0	R0	R1	R0	R0
**AST/ALT POD0(U/L)**	939/954	702/717	17/17	242/152	68/55
**morbidity (*C.D.*)**	no	no	no	Pneumonia(*llla*)	no
**Hospital days post op.**	34	27	40	104	18
**Mortality**	no	no	no	no	no
**adjuvant**	no	yes	yes	no	yes

GnP: Gemcitabine + nab-Paclitaxel, GS: Gemcitabine + S1, CA: Celiac Axis, RHA: right hepatic artery, PHA: proper hepatic artery, SpA: splenic artery, LGA: Left gastric artery, PV: portal vein, SMV: superior mesenteric vein, and Ⅲa: Clavien–Dindo Ⅲa.

## Data Availability

Data are contained within the article.

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
