# Peer review of "Pancreatectomy with Celiac Axis Resection and Reconstruction for Locally Advanced Pancreatic Cancer"

_cancers, 2024, doi:10.3390/cancers16234115_

Round 1

Reviewer 1 Report

Comments and Suggestions for Authors

The article primarily explores the feasibility and safety of conversion surgery (CS) for patients with initially unresectable pancreatic cancer who have responded to chemotherapy. Conversion surgery entails a major arterial resection and reconstruction to achieve a complete radical resection. The paper provides a detailed discussion on the critical aspects of pancreatectomy combined with celiac artery (CA) resection and reconstruction, bifurcating it into resection and arterial reconstruction. It underscores the significance of establishing a "golden view," which facilitates a clear visualization of the abdominal aorta, CA, SMA, IVC, and left renal vein from the abdominal side. Pancreatectomy necessitates arterial reconstruction, and it is imperative to maintain postoperative arterial blood flow, for which tension-free and short bypass should be assured. The article also documents that between 2014 and 2024, sixteen URLA patients underwent CS and required major arterial en bloc resection following pretreatment. Of these, five patients needed CA reconstruction surgery. Pathological findings revealed that four out of the five patients were diagnosed with T4 stage pancreatic cancer. The R0 resection rate stood at 80%, one patient encountered Clavien-Dindo IIIa or higher complications, but there were no fatalities. In conclusion, the article stresses that surgeons must be equipped to safely and reliably execute pancreatectomy requiring simultaneous major arterial resection and reconstruction during pretreatment.

The manuscript can be accepted with minor revisions, which are outlined below.

1. To augment the validity of conclusions drawn, it is advisable to expand the sample size in subsequent studies. By amassing a larger dataset, a more exhaustive assessment of the efficacy and safety of pancreaticoduodenectomy in conjunction with arterial reconstruction can be accomplished.

2. To underscore the benefits of this surgical procedure, it is advisable to juxtapose it with prevailing treatment modalities, including chemotherapy and radiotherapy. Such comparisons will elucidate the surgery's role within the broader therapeutic landscape, offering patients a more holistic array of treatment alternatives.

3. A thorough examination of critical metrics, including postoperative survival rates, disease-free intervals, and complication incidences, is imperative. Such an analysis will provide insights into the surgery's overall efficacy and pinpoint domains needing enhancement.

4. The paper should include a detailed discussion on the conditions that warrant surgical intervention, such as identifying suitable candidates for the procedure and outlining circumstances under which it should be avoided. Additionally, addressing the limitations of the surgery and exploring potential enhancements will contribute to the advancement of the field.

5. The introduction section would benefit from the inclusion of additional background information on pancreatic cancer treatment, encompassing chemotherapy and surgery.

6. It is recommended to cite the following literature:

[1] A. Gu, J. Li, S. Qiu, S. Hao, Z.-Y. Yue, S. Zhai, M.-Y. Li, Y. Liu, Pancreatic cancer environment: From patient-derived models to single-cell omics. Mol. Omics 2024, 20, 220233. DOI: 10.1039/D3MO00250K

[2] Guo J, Wang MF, Zhu Y, Watari F, Xu YH, Chen X. Exploitation of platelets for antitumor drug delivery and modulation of the tumor immune microenvironment. Acta Materia Medica. 2023, 2(2): 172-190. DOI: 10.15212/AMM-2023-0005

[3] Y. Yang, S. Guan, Z. Ou, W. Li, L. Yan, B. Situ, Interdiscip. Med. 2023, 1, e20230013. https://doi.org/10.1002/INMD.20230013

Author Response

Thank you for reviewing my paper.

Comment1:

 To augment the validity of conclusions drawn, it is advisable to expand the sample size in subsequent studies. By amassing a larger dataset, a more exhaustive assessment of the efficacy and safety of pancreaticoduodenectomy in conjunction with arterial reconstruction can be accomplished.

Response1: 

 I agree with the reviewer's opinion. In this paper, we stated in Discussion, "Our study has several limitations associated with the inherent errors and biases of small studies. A large number of trials are recommended to further evaluate the feasibility of pancreatectomy requiring CA resection and reconstruction for URLA pancreatic cancer." However, in the future, we would like to conduct further research and report on increasing the number of cases.

Comment2:

 To underscore the benefits of this surgical procedure, it is advisable to juxtapose it with prevailing treatment modalities, including chemotherapy and radiotherapy. Such comparisons will elucidate the surgery's role within the broader therapeutic landscape, offering patients a more holistic array of treatment alternatives.

Response2:

 Thank you for pointing it out. As you know, in the treatment of URLA, a good prognosis has been reported by performing pre-treatment chemotherapy of 4 to 6 Kurr or more, performing radical surgery after chemoradiotherapy, and adding adjuvant chemotherapy [Reference 1- 3]. As shown in Table 1 in our report, we expect that the prognosis will improve by combining the three treatments of preoperative chemotherapy, surgical therapy, and postoperative chemotherapy.

Comment3:

 A thorough examination of critical metrics, including postoperative survival rates, disease-free intervals, and complication incidences, is imperative. Such an analysis will provide insights into the surgery's overall efficacy and pinpoint domains needing enhancement.

Response3:

 Thank you for your valuable comments. As indicated in Response 1, we would like to increase the number of cases in the future and report on long-term complications, disease-free survival period, and overall survival period.

Comment4:

 The paper should include a detailed discussion on the conditions that warrant surgical intervention, such as identifying suitable candidates for the procedure and outlining circumstances under which it should be avoided. Additionally, addressing the limitations of the surgery and exploring potential enhancements will contribute to the advancement of the field.

Response4:

 Thank you for pointing it out. The applicable patients you pointed out are listed in “2.4.1 indication”. The exclusion criteria were determined to be only patients who did not meet the indications.

Comment5:

 The introduction section would benefit from the inclusion of additional background information on pancreatic cancer treatment, encompassing chemotherapy and surgery.

Response5:

Following your suggestion, I added the following text.

Pancreatic cancer is a biologically highly malignant disease that easily leads to lymph node metastasis, nerve invasion, and distant metastasis even when the tumor is small [1]. Since Fortner [2] reported regional pancreatectomy in 1973 in an effort to improve surgical treatment outcomes, prophylactic extended surgery has been attempted worldwide to improve radical operation.  Although some cases have benefited from this procedure, the 5-year survival rate after resection was less than 15%. It was difficult to improve the prognosis with surgery alone. Since Burris et al. [3] reported the superiority of chemotherapy with gemcitabine hydrochloride alone for unresectable and recurrent pancreatic cancer in 1997, several effective chemotherapy drugs have been published. Furthermore, the development of multi-drug combination therapy has progressed, and fluorouracil and calcium folinate combination therapy (FOLFIRINOX therapy) [4] and gemcitabine hydrochloride + nab-paclitaxel combination therapy [5] have become effective treatments for unresectable and recurrent pancreatic cancer.

Comment6: It is recommended to cite the following literature.

Resonse6:

 I will follow your suggestion and use it as references.

Reviewer 2 Report

Comments and Suggestions for Authors

In this manuscript, the authors present their experience of treating patients with locally advanced pancreatic cancer after effective chemotherapy where pancreatic resection was combined with concomitant major arterial resections and reconstruction.

These are extreme major surgeries requiring proper preoperative assessment, experience and vascular surgery competence. It is true that the number of patients shown is small however technical details are sufficiently described along with relevant figures. The reported results are satisfactory considering these extreme surgical cases.

Some comments to the manuscript include:

Please pay attention that all abbreviations are given in full when first mentioned. This applies also for the abstract and the legend figures. In addition, the abbreviations should be explained in the legends of the figures and should include all the abbreviations shown in the figures (see Fig. 3a).

Probably it is better to report Appleby procedure as gastrectomy-distal pancreatectomy-splenectomy for uniformity and clarity.

Please also pay attention for proper grammar and abbreviations, for example:

patients were underwent CS (Abstract, line 36)

the Para Ao lymph node (line 181)=para-aortic

aspect of A0 (line 200)=Ao

checked by Doppler echocardiography (lines 264-265)=Colored Doppler, Triplex?

patients with SD (line 280) = stable disease

Author Response

Thank you for revising my paper.

Comment1:

Please pay attention that all abbreviations are given in full when first mentioned. This applies also for the abstract and the legend figures. In addition, the abbreviations should be explained in the legends of the figures and should include all the abbreviations shown in the figures (see Fig. 3a).

Response1:

I will make corrections according to the reviewer's instructions.

Comment2:

Probably it is better to report Appleby procedure as gastrectomy-distal pancreatectomy-splenectomy for uniformity and clarity.

Response2:

I will change Appleby operation to gastrectomy-distal pancreatectomy-splenectomy (Appleby procedure).

Comment3:

Please also pay attention for proper grammar and abbreviations, for example: patients were underwent CS (Abstract, line 36) the Para Ao lymph node (line 181)=para-aortic aspect of A0 (line 200)=Ao checked by Doppler echocardiography (lines 264-265)=Colored Doppler, Triplex? patients with SD (line 280) = stable disease

Response3:

I will make corrections according to the reviewer's instructions.

Reviewer 3 Report

Comments and Suggestions for Authors

The article "Pancreatectomy with celiac axis resection and reconstruction for locally advanced pancreatic cancer" presents a single centered case series for pancreatectomy with celiac axis resection and reconstruction for locally advanced pancreatic cancer. The article introduces the pancreatectomy with celiac axis resection, which is a complex procedure when tumors grows into nearby tissues and blood vessels. Tumor cannot be removed with standard pancreatic surgery and a highly complex surgical procedure is performed. The article carefully introduces the complexity and guide the procedure in case by case scenario with minute details and illustration. The article will be beneficial as a reference for future handling of such complex cases. This study can be a baseline for future multicentered larger studies.

The article is well-written covering minute to minute technical aspects. However, few minor changes for improvement of the article are as follows:

1.    If possible, please provide the name of adjuvant therapy mentioned in Table 1.

2.    Please list risk and complications in details with specific examples to the cases mentioned in the article.

3.    If possible, please discuss the possible reasons for shorter median survival time than available in literature apart from selection of patients (T4 pancreatic cancer). This information will help to design/plan the future studies and apply for patient treatment.

4.    Overall, the paper is well written however case specific discussion and comparison with available literature will enhance the quality of the information.

5.     A careful proofread is suggested to avoid minor typos and to introduce abbreviation at their first usage.

Author Response

Thank you for revising my paper.

Comment1:  If possible, please provide the name of adjuvant therapy mentioned in Table 1.

Response1: I will follow your instructions and add it.

Comment2: Please list risk and complications in details with specific examples to the cases mentioned in the article.

Response2:

Following your instructions, I added the following text:

Complications of Clavien–Dindo (CD) IIIa or higher was observed in one patient (pneumonia; case4) (Table1). In other patients, there were no perioperative complications, including pancreatic fistula. There were no deaths.

Comment3:  If possible, please discuss the possible reasons for shorter median survival time than available in literature apart from selection of patients (T4 pancreatic cancer). This information will help to design/plan the future studies and apply for patient treatment.

Response3:

Following your instructions, I added the following text:

Despite the small number of cases, the median survival time was 24.5 months, which was not as good as that reported by other studies showing the superiority of CS [25-27]; however, considering that most of the patients had histopathologically T4 pancreatic cancer, the pretreatment period was short, and the effect of the pretreatment was poor (SD accounting for 80% and 60% normalization ratio of CA19-9), more cases are necessary to discuss the pros and cons of treatment.

Comment4: Overall, the paper is well written however case specific discussion and comparison with available literature will enhance the quality of the information.

Response4: 

We will explain your comments. In the Discussion section, we described the general concept and results of conversion surgery. Furthermore in Response 3, we also described the results of pancreatic resection requiring combined resection and reconstruction of the celiac artery, which is specific to this paper.

Comment5: A careful proofread is suggested to avoid minor typos and to introduce abbreviation at their first usage.

Response5: I will follow your instructions and make the corrections. I have asked a native English proofreader to proofread this paper. I have attached a proofreading certificate.
